# On Convergence and Generalization of Dropout Training

**Poorya Mianjy**
Department of Computer Science
Johns Hopkins University
mianjy@jhu.edu

**Raman Arora**
Department of Computer Science
Johns Hopkins University
arora@cs.jhu.edu

## Abstract

We study dropout in two-layer neural networks with rectified linear unit (ReLU) activations. Under mild overparametrization and assuming that the limiting kernel can separate the data distribution with a positive margin, we show that dropout training with logistic loss achieves $\epsilon$-suboptimality in test error in $O(1/\epsilon)$ iterations.

## 1  Introduction

Dropout is an algorithmic regularization approach that endows deep learning models with excellent generalization ability despite the non-convex nature of the underlying learning problem and the capacity of modern over-parameterized models to over-fit. Introduced by Hinton et al. [2012], Srivastava et al. [2014], dropout involves randomly pruning the network at every iteration of backpropagation by turning off a random subset of hidden nodes. Like many popular algorithmic approaches that emerged as heuristics from practitioners with deep insight into the learning problem, dropout, while extremely successful in practice, lacks a strong theoretical justification, especially from a computational learning theoretic perspective.

Dropout has been successful in several application areas including computer vision [Szegedy et al., 2015], natural language processing [Merity et al., 2017], and speech recognition [Dahl et al., 2013]. Motivated by explaining the empirical success of dropout, and inspired by simple, intuitive nature of the method, numerous works in recent years have focused on understanding its theoretical underpinnings [Baldi and Sadowski, 2013, Wager et al., 2013, Helmbold and Long, 2015, Gal and Ghahramani, 2016, Wei et al., 2020]. Most of these works, however, steer clear from the algorithmic and computational learning aspects of dropout. More precisely, none of the prior work, to the best of our knowledge, yields insights into the runtime of learning using dropout on non-linear neural networks. In this paper, we initiate a study into the iteration complexity of dropout training for achieving $\epsilon$-suboptimality on true error – the misclassification error with respect to the underlying population – in two-layer neural networks with ReLU activations.

We leverage recent advances in the theory of deep learning in over-parameterized settings with extremely (or infinitely) wide networks [Jacot et al., 2018, Lee et al., 2019]. Analyzing two-layer ReLU networks in such a regime has led to a series of exciting results recently establishing that gradient descent (GD) or stochastic gradient descent (SGD) can successfully minimize the empirical error and the true error [Li and Liang, 2018, Du et al., 2019, Daniely, 2017, Zou et al., 2018, Allen-Zhu et al., 2019, Song and Yang, 2019, Arora et al., 2019, Cao and Gu, 2019, Oymak and Soltanolkotabi, 2020]. In a related line of research, several works attribute generalization in over-parametrized settings to the implicit inductive bias of optimization algorithms (through the geometry of local search methods) [Neyshabur et al., 2017]. However, many real-world state-of-the-art systems employ various explicit regularizers, and there is growing evidence that implicit bias may be unable to explain generalization even in a simpler setting of stochastic convex optimization [Dauber et al., 2020]. Here, we extend convergence guarantees and generalization bounds for GD-based methods with explicit

regularization due to dropout. We show that the key insights from analysis of GD-based methods in over-parameterized settings carry over to dropout training.

We summarize our key contributions as follows.

1. We give precise non-asymptotic convergence rates for achieving $\epsilon$-suboptimality in the test error via dropout training in two-layer ReLU networks.

2. We show that dropout training implicitly compresses the network. In particular, we show that there exists a sub-network, i.e., one of the iterates of dropout training, that can generalize as well as any complete network.

3. This work contributes to a growing body of work geared toward a theoretical understanding of GD-based methods for regularized risk minimization in over-parameterized settings.

The rest of the paper is organized as follows. In Section 2, we survey the related work. In Section 3, we formally introduce the problem setup and dropout training, state the key assumptions, and introduce the notation. In Section 4, we give the main results of the paper. In Section 5, we present a sketch of the proofs of our main results – the detailed proofs are deferred to the Appendix. We conclude the paper by providing empirical evidence for our theoretical results in Section 6.

## 2   Related Work

Empirical success of dropout has inspired a series of works aimed at understanding its theoretical underpinnings. Most of these works have focused on explaining the algorithmic regularization due to dropout in terms of *conventional* regularizers. Dropout training has been shown to be similar to *weight decay* in linear regression [Srivastava et al., 2014], in generalized linear models [Wager et al., 2013], and in a PAC-Bayes setting [McAllester, 2013]. These results have recently been extended to multi-variate regression models, where dropout induces a *nuclear norm* penalty in single hidden-layer linear networks [Mianjy et al., 2018] and deep linear networks [Mianjy and Arora, 2019], and a *weighted trace-norm* penalty in matrix completion [Arora et al., 2020]. In a recent work, Wei et al. [2020] characterize explicit regularization due to dropout in terms of the derivatives of the loss and the model, and argue for an implicit regularization effect that stems from the stochasticity in dropout updates.

A parallel strand of research has focused on bounding the generalization gap in dropout training, leveraging tools from uniform convergence. The early work of Wager et al. [2014] showed that under a certain topic model assumption on the data, dropout in linear classification can improve the decay of the excess risk of the empirical risk minimizer. Assuming certain norm-bounds on the weights of the network, the works of Wan et al. [2013], Zhai and Wang [2018], Gao and Zhou [2016] showed that the Rademacher complexity of networks trained by dropout decreases with the dropout rate. Finally, Arora et al. [2020] showed further that the Rademacher complexity can be bounded merely in terms of the explicit regularizer induced by dropout.

Despite the crucial insights provided by the previous art, there is not much known about the non-asymptotic convergence behaviour of dropout training in the literature. A very recent work by Senen-Cerda and Sanders [2020] shows for deep neural networks with polynomially bounded activations with continuous derivatives, under squared loss, that the network weights converge to a stationary set of system of ODEs. In contrast, our results leverages over-parameterization in two-layer networks with non-differentiable ReLU activations, works with logistic loss, and establishes $\epsilon$-suboptimality in the true misclassification error.

Our results are inspired by the recent advances in over-parameterized settings. A large body of literature has focused on deriving optimization theoretic guarantees for (S)GD in this setting. In particular, Li and Liang [2018], Du et al. [2019] were among the first to provide convergence rates for empirical risk minimization using GD. Several subsequent works extended those results beyond two-layers, for smooth activation functions [Du et al., 2018], and general activation functions [Allen-Zhu et al., 2018, Zou et al., 2018] .

Learning theoretic aspects of GD-based methods have been studied for several important target concept classes. Under linear-separability assumption, via a compression scheme, Brutzkus et al. [2017] showed that SGD can efficiently learn a two-layer ReLU network. Li and Liang [2018] further showed that SGD enjoys small generalization error on two-layer ReLU networks if the data follows a

well-separated mixture of distributions. Allen-Zhu et al. [2019] showed generalization error bounds for SGD in two- and three-layer networks with smooth activations where the concept class has fewer parameters. Arora et al. [2019] proved data-dependent generalization error bounds based on the neural tangent kernel by analyzing the Rademacher complexity of the class of networks reachable by GD.

When the data distribution can be well-classified in the *random feature space* induced by the gradient of the network at initialization, Cao and Gu [2019] provide generalization guarantees for SGD in networks with arbitrary depth. Nitanda and Suzuki [2019] studied convergence of GD in two-layer networks with smooth activations, when the data distribution is further *separable* in the infinite-width limit of the random feature space. Ji and Telgarsky [2019] adopted the same margin assumption and improved the convergence rate as well as the over-parameterization size for non-smooth ReLU activations. Here, we generalize the margin assumption in Nitanda and Suzuki [2019] to take into account the randomness injected by dropout into the gradient of the network at initialization, or equivalently, the scale of the corresponding random feature. Our work is most closely related to and inspired by Ji and Telgarsky [2019]; however, we analyze dropout training as opposed to plain SGD, give generalization bounds in expectation, and show the compression benefits of dropout training.

We emphasize that all of the results above focus on (S)GD in absence of any explicit regularization. We summarize a few papers that study regularization in the over-parameterized setting. The work of Wei et al. [2019] showed that even simple explicit regularizers such as weight decay can indeed provably improve the sample complexity of training using GD in the Neural Tangent Kernel (NTK) regime, appealing to a margin-maximization argument in homogeneous networks. We also note the recent works by Li et al. [2019] and Hu et al. [2020], which studied the robustness of GD to noisy labels, with explicit regularization in forms of early stopping; and squared norm of the distance from initialization, respectively.

# 3   Preliminaries and Notation

Let $\mathcal{X} \subseteq \mathbb{R}^d$ and $\mathcal{Y} = \{\pm 1\}$ denote the input and label spaces, respectively. We assume that the data is jointly distributed according to an unknown distribution $\mathcal{D}$ on $\mathcal{X} \times \mathcal{Y}$. Given $T$ i.i.d. examples $\mathcal{S}_T = \{(x_t, y_t)\}_{t=1}^T \sim \mathcal{D}^T$, the goal of learning is to find a hypothesis $f(\cdot; \Theta) : \mathcal{X} \to \mathbb{R}$, parameterized by $\Theta$, that has a small *misclassification error* $\mathcal{R}(\Theta) := \mathbb{P}\{yf(x; \Theta) < 0\}$. Given a convex surrogate loss function $\ell : \mathbb{R} \to \mathbb{R}_{\geq 0}$, a common approach to the above learning problem is to solve the stochastic optimization problem $\min_\Theta L(\Theta) := \mathbb{E}_\mathcal{D}[\ell(yf(x; \Theta))]$.

In this paper, we focus on logistic loss $\ell(z) = \log(1 + e^{-z})$, which is one of the most popular loss functions for classification tasks. We consider two-layer ReLU networks of width $m$, parameterized by the "weights" $\Theta = (W, a) \in \mathbb{R}^{m \times d} \times \mathbb{R}^m$, computing the function $f(\cdot; \Theta) : x \mapsto \frac{1}{\sqrt{m}} a^\top \sigma(Wx)$. We initialize the network with $a_r \sim \text{Unif}(\{+1, -1\})$ and $w_{r,1} \sim \mathcal{N}(0, I)$, for all hidden nodes $r \in [m]$. We then fix the top layer weights and train the hidden layer $W$ using the dropout algorithm. We denote the weight matrix at time $t$ by $W_t$, and $w_{r,t}$ represents its $r$-th column. For the sake of simplicity of the presentation, we drop non-trainable arguments from all functions, e.g., we use $f(\cdot; W)$ in lieu of $f(\cdot; \Theta)$.

Let $B_t \in \mathbb{R}^{m \times m}$, $t \in [T]$, be a random diagonal matrix with diagonal entries drawn independently and identically from a Bernoulli distribution with parameter $q$, i.e., $b_{r,t} \sim \text{Bern}(q)$, where $b_{r,t}$ is the $r$-th diagonal entry of $B_t$. At the $t$-th iterate, dropout entails a SGD step on (the parameters of) the sub-network $g(W; x, B_t) = \frac{1}{\sqrt{m}} a^\top B_t \sigma(Wx)$, yielding updates of the form $W_{t+\frac{1}{2}} \leftarrow W_t - \eta \nabla \ell(y_t g(W_t; x_t, B_t))$. The iteration concludes with projecting the incoming weights – i.e. rows of $W_{t+\frac{1}{2}}$ – onto a pre-specified Euclidean norm ball. We note that such max-norm constraints are standard in the practice of deep learning, and has been a staple to dropout training since it was proposed in Srivastava et al. [2014][1]. Finally, at test time, the weights are multiplied by $q$ so as

**Algorithm 1** Dropout in Two-Layer Networks

---

**Input:** data $\mathcal{S}_T = \{(\mathrm{x}_t, y_t)\}_{t=1}^T \sim \mathcal{D}^T$; Bernoulli masks $\mathcal{B}_T = \{\mathrm{B}_t\}_{t=1}^T$;
    dropout rate $1 - q$; max-norm constraint parameter $c$; learning rate $\eta$

1: *initialize:* $\mathrm{w}_{r,1} \sim \mathcal{N}(0, \mathrm{I})$ and $a_r \sim \mathrm{Unif}(\{+1, -1\})$, $r \in [m]$
2: **for** $t = 1, \dots, T - 1$ **do**
3:     *forward:* $g(\mathrm{W}_t; \mathrm{x}_t, \mathrm{B}_t) = \frac{1}{\sqrt{m}} \mathrm{a}^\top \mathrm{B}_t \sigma(\mathrm{W}_t \mathrm{x}_t)$
4:     *backward:* $\nabla L_t(\mathrm{W}_t) = \nabla \ell(y_t g(\mathrm{W}_t; \mathrm{x}_t, \mathrm{B}_t)) = \ell'(y_t g(\mathrm{W}_t; \mathrm{x}_t, \mathrm{B}_t)) \cdot y_t \nabla g(\mathrm{W}_t; \mathrm{x}_t, \mathrm{B}_t)$
5:     *update:* $\mathrm{W}_{t+\frac{1}{2}} \leftarrow \mathrm{W}_t - \eta \nabla L_t(\mathrm{W}_t)$
6:     *max-norm:* $\mathrm{W}_{t+1} \leftarrow \Pi_c(\mathrm{W}_{t+\frac{1}{2}})$
7: **end for**

**Test Time:** re-scale the weights as $\mathrm{W}_t \leftarrow q\mathrm{W}_t$

---

to make sure that the output at test time is on par with the expected output at training time. The pseudo-code for dropout training is given in Algorithm 1[2].

Our analysis is motivated by recent developments in understanding the dynamics of (S)GD in the so-called *lazy regime*. Under certain initialization, learning rate, and network width requirements, these results show that the iterates of (S)GD tend to stay close to initialization; therefore, a first-order Taylor expansion of the $t$-th iterate around initialization, i.e. $f(\mathrm{x}; \mathrm{W}_t) \approx f(\mathrm{x}; \mathrm{W}_1) + \langle \nabla f(\mathrm{x}; \mathrm{W}_1), \mathrm{W}_t - \mathrm{W}_1 \rangle$, can be used as a proxy to track the evolution of the network predictions [Li and Liang, 2018, Chizat et al., 2018, Du et al., 2019, Lee et al., 2019]. In other words, training in lazy regime reduces to finding a linear predictor in the reproducing kernel Hilbert space (RKHS) associated with the gradient of the network at initialization, $\nabla f(\cdot; \mathrm{W}_1)$. In this work, following Nitanda and Suzuki [2019], Ji and Telgarsky [2019], we assume that the data distribution is separable by a positive margin in the limiting RKHS:

**Assumption 1** $((q, \gamma)$-Margin)**.** Let $\mathrm{z} \sim \mathcal{N}(0, \mathrm{I}_d)$ and $b \sim \mathrm{Bern}(q)$ be a $d$-dimensional standard normal random vector, and a Bernoulli random variable with parameter $q$, respectively. There exists a *margin parameter* $\gamma > 0$, and a linear transformation $\psi : \mathbb{R}^d \to \mathbb{R}^d$ satisfying A) $\mathbb{E}_{\mathrm{z}}[\|\psi(\mathrm{z})\|^2] < \infty$; B) $\|\psi(\mathrm{z})\|_2 \leq 1$ for all $\mathrm{z} \in \mathbb{R}^d$; and C) $\mathbb{E}_{\mathrm{z},b}[y \langle \psi(\mathrm{z}), b\mathrm{x} \mathbb{I}[\mathrm{z}^\top \mathrm{x} \geq 0] \rangle] \geq \gamma$ for almost all $(\mathrm{x}, y) \sim \mathcal{D}$.

The above assumption provides an infinite-width extension to the separability of data in the RKHS induced by $\nabla g(\mathrm{W}_1; \cdot, \mathrm{B}_1)$. To see that, define $\mathrm{V} := [\mathrm{v}_1, \dots, \mathrm{v}_m]^\top \in \mathbb{R}^{m \times d}$, where $\mathrm{v}_r = \frac{1}{\sqrt{m}} a_r \psi(\mathrm{w}_{r,1})$ for all $r \in [m]$, satisfying $\|\mathrm{V}\|_F \leq 1$. For any given point $(\mathrm{x}, y) \in \mathcal{X} \times \mathcal{Y}$, the margin attained by $\mathrm{V}$ is at least $y \langle \nabla g(\mathrm{W}_1; \mathrm{x}, \mathrm{B}_1), \mathrm{V} \rangle = \frac{1}{m} \sum_{r=1}^m y \langle \psi(\mathrm{w}_{r,1}), b_{r,1} \mathrm{x} \mathbb{I}\{\mathrm{w}_{r,1}^\top \mathrm{x} > 0\} \rangle$, which is a finite-width approximation of the quantity $\mathbb{E}[y \langle \psi(\mathrm{z}), b\mathrm{x} \mathbb{I}\{\mathrm{z}^\top \mathrm{x} > 0\} \rangle]$ in Assumption 1.

We remark that when $q = 1$ (pure SGD – no dropout), with probability one it holds that $b = 1$, so that Assumption 1 boils down to that of Nitanda and Suzuki [2019] and Ji and Telgarsky [2019]. When $q < 1$, this assumption translates to a margin of $\gamma/q$ on the *full* features $\nabla f(\cdot; \mathrm{W}_1)$, which is the appropriate scaling given that $\nabla f(\cdot; \mathrm{W}_1) = \frac{1}{q} \mathbb{E}_{\mathrm{B}}[\nabla g(\mathrm{W}_1; \cdot, \mathrm{B})]$. Alternatively, dropout training eventually outputs a network with weights scaled down as $q\mathrm{W}_t$, which (in expectation) corresponds to the shrinkage caused by the Bernoulli mask in $b\mathrm{x} \mathbb{I}\{\mathrm{z}^\top \mathrm{x} > 0\}$. Regardless, we note that *our analysis can be carried over even without this scaling*; however, new polynomial factors of $1/q$ will be introduced in the required width in our results in Section 4.

## 3.1 Notation

We denote matrices, vectors, scalar variables and sets by Roman capital letters, Roman small letters, small letters, and script letters, respectively (e.g. Y, y, $y$, and $\mathcal{Y}$). The $r$-th entry of vector y, and the $r$-th row of matrix Y, are denoted by $y_i$ and $\mathrm{y}_i$, respectively. Furthermore, for a sequence of matrices $\mathrm{W}_t, t \in \mathbb{N}$, the $r$-th row of the $t$-th matrix is denoted by $\mathrm{w}_{r,t}$. Let $\mathbb{I}$ denote the indicator of an event, i.e., $\mathbb{I}\{y \in \mathcal{Y}\}$ is one if $y \in \mathcal{Y}$, and zero otherwise. For any integer $d$, we represent the set $\{1, \dots, d\}$ by $[d]$. Let $\|\mathrm{x}\|$ represent the $\ell_2$-norm of vector x; and $\|\mathrm{X}\|$, and $\|\mathrm{X}\|_F$ represent the

operator norm, and the Frobenius norm of matrix X, respectively. $\langle \cdot, \cdot \rangle$ represents the standard inner product, for vectors or matrices, where $\langle X, X' \rangle = \mathrm{Tr}(X^\top X')$. For a matrix $W \in \mathbb{R}^{m \times d}$, and a scalar $c > 0$, $\Pi_c(W)$ projects the rows of $W$ onto the Euclidean ball of radius $c$ with respect to the $\ell_2$-norm.

For any $t \in [T]$ and any $W$, let $f_t(W) := f(x_t; W)$ denote the network output given input $x_t$, and let $g_t(W) := g(W; x_t, B_t)$ denote the corresponding output of the sub-network associated with the dropout pattern $B_t$. Let $L_t(W) = \ell(y_t g_t(W))$ and $Q_t(W) = -\ell'(y_t g_t(W))$ be the associated instantaneous loss and its negative derivative. The partial derivative of $g_t$ with respect to the $r$-th hidden weight vector is given by $\frac{\partial g_t(W)}{\partial w_r} = \frac{1}{\sqrt{m}} a_r b_{r,t} \mathbb{I}\{w_r^\top x_t \geq 0\} x_t$. We denote the linearization of $g_t(W)$ based on features at time $t$ by $g_t^{(k)}(W) := \langle \nabla g_t(W_k), W \rangle$; and its corresponding instantaneous loss and its negative derivative by $L_t^{(k)}(W) := \ell(y_t g_t^{(k)}(W))$ and $Q_t^{(k)}(W) := -\ell'(y_t g_t^{(k)}(W))$, respectively. $Q$ plays an important role in deriving generalization bounds for dropout sub-networks $g(W_t; x, B_t)$; it has been recently used in [Cao and Gu, 2019, Ji and Telgarsky, 2019] for analyzing the convergence of SGD and bounding its generalization error.

We conclude this section by listing a few useful identities that are used throughout the paper. First, due to homogeneity of the ReLU, it holds that $g_t^{(t)}(W_t) = \langle \nabla g_t(W_t), W_t \rangle = g_t(W_t)$. Moreover, the norm of the network gradient, and the norm of the the gradient of the instantaneous loss can be upper-bounded as $\|\nabla g_t(W)\|_F^2 = \sum_{r=1}^m \|\frac{\partial g_t(W)}{\partial w_r}\|^2 \leq \frac{\|B_t\|_F^2}{m} \leq 1$, and $\|\nabla L_t(W)\|_F = -\ell'(y_t g_t(W))\|y_t \nabla g_t(W)\|_F \leq Q_t(W)$, respectively. Finally, the logistic loss satisfies $|\ell'(z)| \leq \ell(z)$, so that $Q_t(W) \leq L_t(W)$.

# 4 Main Results

We begin with a simple observation that given the random initialization scheme in Algorithm 1, the $\ell_2$-norm of the rows of $W_1$ are expected to be concentrated around $\sqrt{d}$. In fact, using Gaussian concentration inequality (Theorem A.1 in the appendix), it holds with probability at least $1 - 1/m$, uniformly for all $r \in [m]$, that $\|w_{r,1}\| \leq \sqrt{d} + 2\sqrt{\ln(m)}$. For the sake of the simplicity of the presentation, we assume that the event $\max_{r \in [m]} \|w_{r,1}\| \leq 2\sqrt{\ln(m)}$ holds through the run of dropout training. Alternatively, we can re-initialize the weights until this condition is satisfied, or multiply the probability of success in our theorems by a factor of $1 - 1/m$.

Our first result establishes that the true misclassification error of dropout training vanishes as $\tilde{\mathcal{O}}(1/T)$.

**Theorem 4.1** (Learning with Dropout). Let $c = \sqrt{d} + \max\{\frac{1}{14\gamma^2}, 2\sqrt{\ln(m)}\} + 1$ and $\lambda := 5\gamma^{-1}\ln(2\eta T) + \sqrt{44\gamma^{-2}\ln(24\eta c\sqrt{m}T^2)}$. Under Assumption 1, for any learning rate $\eta \in (0, \ln(2)]$ and any network of width satisfying $m \geq 2401\gamma^{-6}\lambda^2$, with probability one over the randomization due to dropout, we have that

$$\min_{t \in [T]} \mathbb{E}[\mathcal{R}(qW_t)] \leq \frac{1}{T}\sum_{t=1}^T \mathbb{E}[\mathcal{R}(qW_t)] \leq \frac{4\lambda^2}{\eta T} = \mathcal{O}\left(\frac{\ln(T)^2 + \ln(mdT)}{T}\right),$$

where the expectation is with respect to the initialization and the training samples.

Theorem 4.1 shows that dropout successfully trains the complete network $f(\cdot; W_t)$. Perhaps more interestingly, our next result shows that dropout successfully trains a potentially significantly narrower sub-network $g(W_t; \cdot, B_t)$. For this purpose, denote the misclassification error due to a network with weights $W$ given a Bernoulli mask $B$ as follows

$$\mathcal{R}(W; B) := \mathbb{P}\{yg(W; x, B) < 0\}.$$

Then the following result holds for the misclassification error of the iterates of dropout training.

**Theorem 4.2** (Compression with Dropout). Under the setting of Theorem 4.1, with probability at least $1 - \delta$ over initialization, the training data, and the randomization due to dropout, we have that

$$\min_{t \in [T]} \mathcal{R}(W_t; B_t) \leq \frac{1}{T}\sum_{t=1}^T \mathcal{R}(W_t; B_t) \leq \frac{12\lambda^2}{\eta T} + \frac{6\ln(1/\delta)}{T} = \mathcal{O}\left(\frac{\ln(mT)}{T}\right).$$

A few remarks are in order.

Theorem 4.1 gives a generalization error bound in expectation. A technical challenge here stems from the unboundedness of the logistic loss. In our analysis, the max-norm constraint in Algorithm 1 is essential to guarantee that the logistic loss remains bounded through the run of the algorithm, thereby controlling the loss of the iterates in expectation. However, akin to analysis of SGD in the lazy regime, the iterates of dropout training are not likely to leave a small proximity of the initialization whatsoever. Therefore, for the particular choice of $c$ in the above theorems, the max-norm projections in Algorithm 1 will be virtually inactive for a typical run.

The expected width of the sub-networks in Theorem 4.2 is only $qm$. Using Hoeffding's inequality and a union bound argument, for any $\delta \in (0,1)$, with probability at least $1 - \delta$, it holds for all $t \in [T]$ that $g(\mathrm{W}_t; \mathrm{x}, \mathrm{B}_t)$ has at most $qm + \sqrt{2m \ln(T/\delta)}$ active hidden neurons. That is, in a typical run of the dropout training, with high probability, there exists a sub-network of width $\approx qm + \tilde{\mathcal{O}}(\sqrt{m})$ whose generalization error is no larger than $\tilde{\mathcal{O}}(1/T)$. In Section 6, we further provide empirical evidence to verify this compression result. We note that dropout has long been considered as a means of network compression, improving post-hoc pruning [Gomez et al., 2019], in Bayesian settings [Molchanov et al., 2017], and in connection with the Lottery Ticket Hypothesis [Frankle and Carbin, 2019]. However, we are not aware of any theoretical result supporting that claim prior to our work.

Finally, the sub-optimality results in both Theorem 4.1 and Theorem 4.2 are agnostic to the dropout rate $1 - q$. This is precisely due to the $(q, \gamma)$-Margin assumption: if it holds, then so does $(q', \gamma)$-Margin for any $q' \in [q, 1]$. That is, these theorems hold for *any* dropout rate not exceeding $1 - q$. Therefore, in light of the remark above, larger admissible dropout rates are preferable since they result in higher compression rates, while enjoying the same generalization error guarantees.

# 5   Proofs

We begin by bounding $\mathbb{E}_{\mathcal{S}_t}[\mathcal{R}(q\mathrm{W}_t)]$, the expected population error of the iterates, in terms of $\mathbb{E}_{\mathcal{S}_t, \mathcal{B}_t}[L_t(\mathrm{W}_t)]$, the expected instantaneous loss of the random sub-networks. In particular, using simple arguments including the smoothing property, the fact that $\mathrm{W}_t$ is independent from $(\mathrm{x}_t, y_t)$ given $\mathcal{S}_{t-1}$, and that logistic loss upper-bounds the zero-one loss, it is easy to bound the expected population risk as $\mathbb{E}_{\mathcal{S}_t}[\mathcal{R}(q\mathrm{W}_t)] \leq \mathbb{E}_{\mathcal{S}_t}[\ell(y_t f(\mathrm{x}_t; q\mathrm{W}_t))]$. Furthermore, using Jensen's inequality, we have that $\ell(y_t f_t(q\mathrm{W}_t)) \leq \mathbb{E}_{\mathrm{B}_t}[L_t(\mathrm{W}_t)]$. The upper bound then follows from these two inequalities. In the following, we present the main ideas in bounding the average instantaneous loss of the iterates.

Under Algorithm 1, dropout iterates are guaranteed to remain in the set $\mathcal{W}_c := \{\mathrm{W} \in \mathbb{R}^{m \times d} : \|\mathrm{w}_r\| \leq c\}$. Using this property and the dropout update rule, we track the distance of consecutive iterates $(\mathrm{W}_{t+1}, \mathrm{W}_t)$ from any competitor $\mathrm{U} \in \mathcal{W}_c$, which leads to the following upper bound on the average instantaneous loss of iterates.

**Lemma 5.1.** Let $\mathrm{W}_1, \ldots, \mathrm{W}_T$ be the sequence of dropout iterates with a learning rate satisfying $\eta \leq \ln(2)$. Then, it holds for any $\mathrm{U} \in \mathcal{W}_c$ that

$$\frac{1}{T} \sum_{t=1}^{T} L_t(\mathrm{W}_t) \leq \frac{\|\mathrm{W}_1 - \mathrm{U}\|_F^2}{\eta T} + \frac{2}{T} \sum_{t=1}^{T} L_t^{(t)}(\mathrm{U}). \tag{1}$$

Note that the upper bound in Equation (1) holds for *any* competitor $\mathrm{U} \in \mathcal{W}_c$; however, we seek to minimize the upper-bound on the right hand side of Equation (1) by finding a *sweet spot* that maintains a trade-off between 1) the distance from initialization, and 2) the average instantaneous loss for the linearized models. Following Ji and Telgarsky [2019], we represent such a competitor as an interpolation between the initial weights $\mathrm{W}_1$ and the *max-margin* competitor $\mathrm{V}$, i.e. $\mathrm{U} := \mathrm{W}_1 + \lambda \mathrm{V}$, where $\lambda$ is the trade-off parameter. Recall that $\mathrm{V} := [\mathrm{v}_1, \cdots, \mathrm{v}_m] \in \mathbb{R}^{d \times m}$, where $\mathrm{v}_r = \frac{1}{\sqrt{m}} a_r \psi(\mathrm{w}_{r,1})$ for any $r \in [m]$, and $\psi$ is given by assumption 1 . Thus, the first term on the right hand side above can be conveniently bounded as $\frac{\lambda^2}{\eta T}$; Lemma 5.2 bounds the second term as follows.

**Lemma 5.2.** Under the setting of Theorem 4.1, it holds with probability at least $1 - \delta$ simultaneously for all iterates $t \in [T]$ that 1) $\|\mathrm{w}_{r,t} - \mathrm{w}_{r,1}\| \leq \frac{7\lambda}{2\gamma\sqrt{m}}$, for all $r \in [m]$; and 2) $L_t^{(t)}(\mathrm{U}) \leq \frac{\lambda^2}{2\eta T}$.

Therefore, left hand side of Equation (1), i.e., the average instantaneous loss of the iterates, can be bounded with high probability as $\frac{2\lambda^2}{\eta T}$. To get the bound in expectation, as presented in Theorem 4.1,

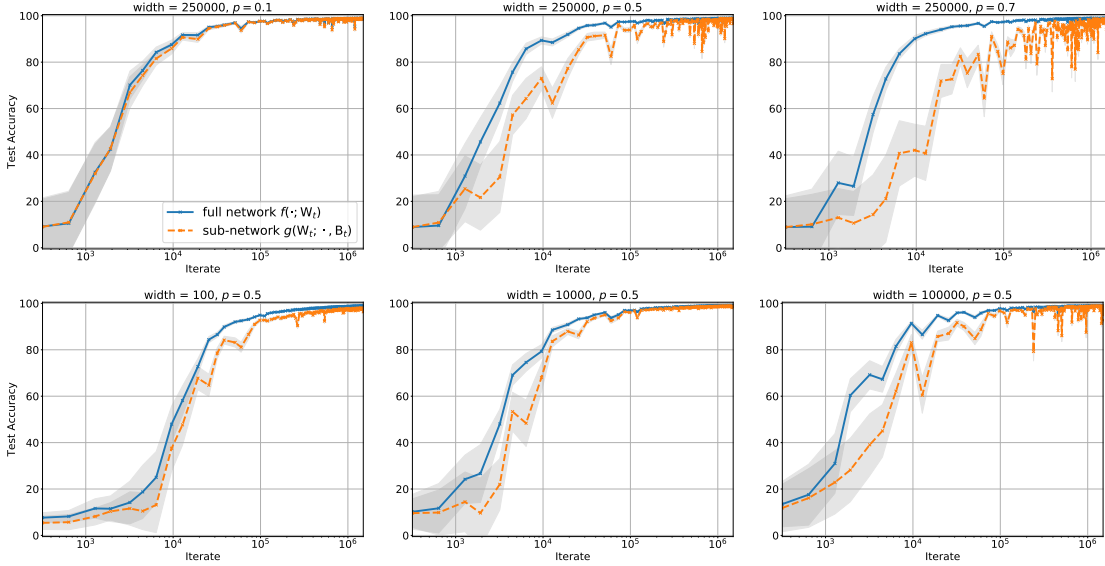

Figure 1: Test accuracy of the full network $f(\cdot; W_t)$ as well as the sub-networks $g(W_t; \cdot, B_t)$ drawn by dropout iterates, as a function of number of iterations $t$, for (**top**) fixed width $m = 250K$ and several dropout rates $1-p \in \{0.1, 0.5, 0.7\}$; and (**bottom**) fixed dropout rate $1-p = 0.5$ and several widths $m \in \{100, 10K, 100K\}$.

we need to control $L_t^{(t)}(U)$ in worst-case scenario. We take advantage of the max-norm constraints in Algorithm 1, and show in Lemma B.5 that with probability one all iterates satisfy $L_t^{(t)}(U) \leq \frac{c\sqrt{m}}{\ln(2)} + 1$. The proof of Theorem 4.1 then follows from carefully choosing $\delta$, the confidence parameter. To prove the compression result in Theorem 4.2, we use the fact that the zero-one loss can be bounded in terms of the negative derivative of the logistic loss [Cao and Gu, 2019]. Therefore, we can bound $\mathcal{R}(W_t; B_t)$, the population risk of the sub-networks, in terms of $Q(W_t; B_t) = \mathbb{E}_{\mathcal{D}}[-\ell'(y_t g_t(W_t))]$. Following Ji and Telgarsky [2019], the proof of Theorem 4.2 then entails showing that $\sum_{t=1}^{T} Q(W_t; B_t)$ is close to $\sum_{t=1}^{T} Q_t(W_t)$, which itself is bounded by the average instantaneous loss of the iterates.

We now present the main ideas in proving Lemma 5.2, which closely follows Ji and Telgarsky [2019]. Since $L_t^{(t)}(U) \leq e^{-y_t \langle \nabla g_t(W_t), U \rangle}$, the proof entails lower bounding $y_t \langle \nabla g_t(W_t), U \rangle$, which can be decomposed as follows

$$y_t \langle \nabla g_t(W_t), U \rangle = y_t \langle \nabla g_t(W_1), W_1 \rangle + y_t \langle \nabla g_t(W_t) - \nabla g_t(W_1), W_1 \rangle \\ + \lambda y_t \langle \nabla g_t(W_1), V \rangle + \lambda y_t \langle \nabla g_t(W_t) - \nabla g_i(B_t W_1), V \rangle. \quad (2)$$

By homogeneity of the ReLU activations, the first term in Equation (2) precisely computes $y_t g_t(W_1)$, which cannot be too negative under the initialization scheme used in Algorithm 1, as we show in Lemma B.3. On the other hand, we show in Lemma B.4 that under Assumption 1, V has a good margin with respect to the randomly initialized weights $W_1$, so that the third term in Equation (2) is concentrated around the margin parameter $\gamma$. The second and the fourth terms in Equation (2) can be bounded thanks to the lazy regime, where $W_t$ remains close to $W_1$ at all times. In particular, provided $\|w_{r,t} - w_{r,1}\| \leq \frac{7\lambda}{2\gamma\sqrt{m}}$, we show in Lemma B.1 that at most only $O(1/\sqrt{m})$-fraction of neural activations change, and thus $\nabla g_t(W_t) - \nabla g_t(W_1)$ has a small norm. Lemma 5.2 then follows from carefully choosing $\lambda$ and $m$ such that the right hand side of Equation (2) is sufficiently positive.

# 6 Experimental Results

The goal of this section is to investigate if dropout indeed compresses the model, as predicted by Theorem 4.2. In particular, we seek to understand if the (sparse) dropout sub-networks $g(W; \cdot, B)$ – regardless of being visited by dropout during the training – are comparable to the full network $f(\cdot; W)$, in terms of the test accuracy.

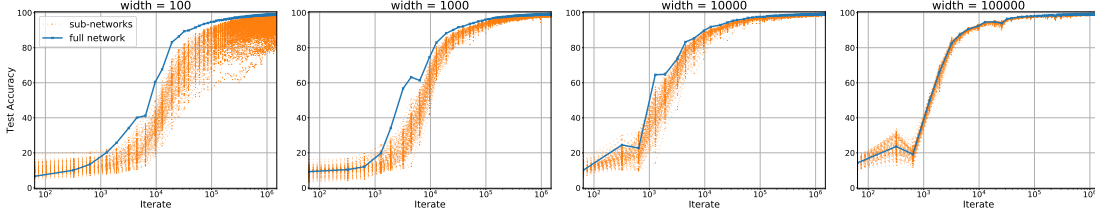

Figure 2: Test accuracy of the full network $f(\cdot; W_t)$ as well as 100 random sub-networks $g(W_t; \cdot, B_1), \ldots, g(W_t; \cdot, B_{100})$ with dropout rate $1 - p = 0.8$, as a function of number of iterations $t$, for several width $m \in \{100, 1K, 10K, 100K\}$.

We train a convolutional neural network with a dropout layer on the top hidden layer, using cross-entropy loss, on the MNIST dataset. The network consists of two convolutional layers with max-pooling, followed by three fully-connected layers. All the activations are ReLU. We use a constant learning rate $\eta = 0.01$ and batch-size equal to $64$ for all the experiments. We train several networks where except for the top layer widths ($m \in \{100, 500, 1K, 5K, 10K, 50K, 100K, 250K\}$), all other architectural parameters are fixed. We track the test accuracy over 25 epochs as a function of number of iterations, for the full network, the sub-networks visited by dropout, as well as random but fixed sub-networks that are drawn independently, using the same dropout rate. We run the experiments for several values of the dropout rate, $1 - p \in \{0.1, 0.2, 0.3, \ldots, 0.9\}$.

Figure 1 shows the test accuracy of the full network $f(\cdot; W_t)$ (blue, solid curve) as well as the dropout iterates $g(W_t; \cdot, B_t)$ (orange, dashed curve), as a function of the number of iterations. Both curves are averaged over 50 independent runs of the experiment; the grey region captures the standard deviation. It can be seen that the (sparse) sub-networks drawn by dropout during the training, are indeed comparable to the full network in terms of the generalization error. As expected, the gap between the full network and the sparse sub-networks is higher for narrower networks, and for higher dropout rates. This figure verifies our compression result in Theorem 4.2.

Next, we show that dropout also generalizes to sub-networks that were not observed during the training. In other words, random sub-networks drawn from the same Bernoulli distribution, also performed well. We run the following experiment on the same convolutional network architecture described above with widths $m \in \{100, 1K, 10K, 100K\}$. We draw 100 sub-networks $g(W; \cdot, B_1), \ldots, g(W; \cdot, B_{100})$, corresponding to diagonal Bernoulli matrices $B_1, \ldots, B_{100}$, all generated by the same Bernoulli distribution used at training (Bernoulli parameter $p = 0.2$, i.e., dropout rate $1 - p = 0.8$). In Figure 2, we plot the generalization error of these sub-networks as well as the full network as a function of iteration number, as orange and blue curves, respectively. We observe that, as the width increases, the sub-networks become increasingly more competitive; it is remarkable that the effective width of these *typical* sub-networks are only $\approx 1/5$ of the full network.

## 7   Conclusion

Most of the results in the literature of over-parameterized neural networks focus on GD-based methods without any explicit regularization. On the other hand, recent theoretical investigations have challenged the view that *implicit bias* due to such GD-based methods can explain generalization in deep learning [Dauber et al., 2020]. Therefore, it seems crucial to explore algorithmic regularization techniques in over-parameterized neural networks. This paper takes a step towards understanding a popular algorithmic regularization technique in deep learning. In particular, assuming that the data distribution is separable in the RKHS induced by the neural tangent kernel, this paper presents precise iteration complexity results for dropout training in two-layer ReLU networks using the logistic loss.

## Broader Impact

We investigate the convergence and generalization of a popular algorithmic regularization technique in deep learning. Although we can not think of any direct social impacts per se, we hope such theoretical studies serve the community in long-term, by improving our understanding of the foundations, which shall eventually lead to more powerful machine learning systems.

## Acknowledgements

This research was supported, in part, by NSF BIGDATA award IIS-1546482, NSF CAREER award IIS-1943251 and NSF TRIPODS award CCF-1934979. Poorya Mianjy acknowledges support as a MINDS fellow. Raman Arora acknowledges support from the Simons Institute as part of the program on the Foundations of Deep Learning and the Institute for Advanced Study (IAS), Princeton, New Jersey, as part of the special year on Optimization, Statistics, and Theoretical Machine Learning.

## Footnotes

[1]Quote from Srivastava et al. [2014]: "One particular form of regularization was found to be especially useful for dropout— constraining the norm of the incoming weight vector at each hidden unit to be upper bounded by a fixed constant $c$"

[2]In a popular variant that is used in machine learning frameworks such as PyTorch, known as inverted dropout, (inverse) scaling is applied at the training time instead of the test time. The inverted dropout is equivalent to the method we study here, and can be analyzed in a similar manner.

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
