[Supplementary Material]

# "On Convergence and Generalization of Dropout Training"

## A  Auxiliary Theorems

**Theorem A.1** (Gaussian Concentration Vershynin [2018]). Consider a random vector $z \sim \mathcal{N}(0, I_d)$ and a $\rho$-Lipschitz function $f : \mathbb{R}^d \to \mathbb{R}$ (with respect to the Euclidean metric). Then $f(z)$ is $\rho$-sub-Gaussian and it holds for all $t \geq 0$:

$$\mathbb{P}\{f(z) - \mathbb{E}[f(z)] \geq t\} \leq e^{\frac{-t^2}{2\rho^2}}$$

**Theorem A.2** (Hoeffding's inequality Vershynin [2018]). Let $X_1, \ldots, X_n$ be independent, mean zero random variables. Assume that $X_i \in [m_i, M_i]$ for every $i$. Then, for every $t > 0$, we have

$$\mathbb{P}\{\sum_{i=1}^{n} X_i \geq t\} \leq e^{-\frac{2t^2}{\Sigma_{i=1}^{n}(m_i - M_i)^2}}$$

**Theorem A.3** (Theorem 1 of Beygelzimer et al. [2011]). Let $X_1, \ldots, X_T$ be a sequence of real-valued random variables. Let $\mathbb{E}_t[Y] := \mathbb{E}[Y|X_1, \ldots, X_{t-1}]$. Assume, for all $t$, that $X_t \leq R$ and that $\mathbb{E}_t[X_t] = 0$. Define the random variable $S_t := \sum_{k=1}^{t} X_t$, and $V_t := \sum_{k=1}^{t} \mathbb{E}_k[X_k^2]$. Then for any $\delta > 0$, with probability at least $1 - \delta$, we have the following guarantee:

$$S_t \leq R \ln\left(\frac{1}{\delta}\right) + (e - 2)\frac{V_t}{R}$$

## B  Proofs

The following Lemma bounds $|R_t|$, where $R_t := \{r \in [m]| \; \mathbb{I}\{w_{r,t}^\top x_t > 0\} \neq \mathbb{I}\{w_{r,1}^\top x_t > 0\}\}$ is the set of hidden nodes at time $t$ whose activation on sample $x_t$ is different from the initialization.

**Lemma B.1.** Assume that $\|w_{r,1} - w_{r,t}\| \leq D$ holds for all $r \in [m]$, where $D$ is a positive constant. Then, with probability at least $1 - \frac{\delta}{3}$, we have that

$$|R_t| \leq mD + \sqrt{\frac{m \ln(3T/\delta)}{2}}, \text{ for all } t \in [T].$$

*Proof of Lemma B.1.* Assume that $r \in R_t$. Then it holds that

$$
\begin{aligned}
|w_{r,1}^\top x_t| &\leq |w_{r,1}^\top x_t| + |w_{r,t}^\top x_t| \\
&= |(w_{r,1} - w_{r,t})^\top x_t| && (r \in R_t) \\
&\leq \|w_{r,1} - w_{r,t}\|\|x_t\| && \text{(Cauchy-Schwarz)} \\
&= \|w_{r,1} - w_{r,t}\| \leq D && (\|x_t\| = 1)
\end{aligned}
$$

Since $w_{r,1}^\top x_t$ is a standard Gaussian random variable, by anti-concentration property of the Gaussian distribution, $\mathbb{E}[\mathbb{I}\{|w_{r,1}^\top x_t| \leq D\}] = \Pr\{|w_{r,1}^\top x_t| \leq D\} \leq \frac{2D}{\sqrt{2\pi}}$. On the other hand, we have that

$$|R_t| = \left|\{r| \; \mathbb{I}\{w_{r,t}^\top x_t > 0\} \neq \mathbb{I}\{w_{r,1}^\top x_t > 0\}\}\right| \leq |\{r| \; |w_{r,1}^\top x_t| \leq D\}| = \sum_{r=1}^{m} \mathbb{I}\{|w_{r,1}^\top x_t| \leq D\}$$

By Hoeffding's inequality, we have the following with probability at least $1 - \frac{\delta}{3T}$:

$$\frac{1}{m}\sum_{r=1}^{m} \mathbb{I}\{|w_{r,1}^\top x_t| \leq D\} \leq \Pr\{|w_{r,1}^\top x_t| \leq D\} + \sqrt{\frac{\ln(3T/\delta)}{2m}} \leq \frac{2D}{\sqrt{2\pi}} + \sqrt{\frac{\ln(3T/\delta)}{2m}}.$$

Multiplying both sides by $m$ and applying union bound on $t \in [T]$ completes the proof. $\square$

**Lemma B.2.** For any $t \in [T]$, let $\mathcal{B}_t := \{B_1, \ldots, B_t\}$ denote the set of Bernoulli masks up to time $t$. Then it holds almost surely that:

$$\sum_{t=1}^{T} \ell(y_t f_t(q\mathrm{W}_t)) \leq \mathbb{E}_{\mathcal{B}_T}[\sum_{t=1}^{T} L_t(\mathrm{W}_t)]. \tag{3}$$

*Proof of Lemma B.2.* For any $a, b \in \mathbb{R}$, the function $\ell(z) = \log(1 + \exp(az + b))$ is convex in $z$. We have the following inequalities:

$$
\begin{aligned}
\mathbb{E}_{\mathcal{B}_T}[\sum_{t=1}^{T} L_t(\mathrm{W}_t)] &= \sum_{t=1}^{T} \mathbb{E}_{\mathcal{B}_t}[\ell(y_t \cdot \frac{1}{\sqrt{m}} \mathrm{a}^\top \mathrm{B}_t \sigma(\mathrm{W}_t \mathrm{x}_t))] \\
&= \sum_{t=1}^{T} \mathbb{E}_{\mathcal{B}_{t-1}}[\mathbb{E}_{\mathrm{B}_t} \ell(y_t \cdot \frac{1}{\sqrt{m}} \sum_{r=1}^{m} a_r b_{r,t} \sigma(\mathrm{w}_{r,t}^\top \mathrm{x}_t)) | \mathcal{B}_{t-1}] \quad \text{(smoothing property)} \\
&\geq \sum_{t=1}^{T} \mathbb{E}_{\mathcal{B}_{t-1}}[\ell(y_t \cdot \frac{1}{\sqrt{m}} \sum_{r=1}^{m} a_r \mathbb{E}_{\mathrm{B}_t}[b_{r,t}] \sigma(\mathrm{w}_{r,t}^\top \mathrm{x}_t)) | \mathcal{B}_{t-1}] \quad \text{(Jensen's inequality)} \\
&= \sum_{t=1}^{T} \ell(y_t \cdot \frac{1}{\sqrt{m}} \sum_{r=1}^{m} a_r \sigma(q\mathrm{w}_{r,t}^\top \mathrm{x}_t)) \quad \quad (\mathbb{E}[b_{r,t}] = q, \text{homogeneity of ReLU}) \\
&= \sum_{t=1}^{T} \ell(y_t f_t(q\mathrm{W}_t))
\end{aligned}
$$

which completes the proof. $\qquad\qquad\qquad\qquad\qquad\qquad\qquad\qquad\qquad\qquad\qquad\qquad\qquad \square$

*Proof of Lemma 5.1.* Using the dropout update rule in Algorithm 1, we start by analyzing the distance of consecutive iterates from the reference point U, assuming that $\Pi_c(\mathrm{U}) = \mathrm{U}$:

$$
\begin{aligned}
\|\mathrm{W}_{t+1} - \mathrm{U}\|_F^2 &= \|\Pi_c(\mathrm{W}_{t+\frac{1}{2}}) - \mathrm{U}\|_F^2 \\
&\leq \|\mathrm{W}_{t+\frac{1}{2}} - \mathrm{U}\|_F^2 \quad\quad\quad\quad\quad\quad\quad\quad\quad\quad (\mathrm{U} \in \mathcal{W}_c) \\
&= \|\mathrm{W}_t - \eta \nabla L_t(\mathrm{W}_t) - \mathrm{U}\|_F^2 \\
&= \|\mathrm{W}_t - \mathrm{U}\|_F^2 - 2\eta \langle \nabla L_t(\mathrm{W}_t), \mathrm{W}_t - \mathrm{U} \rangle + \eta^2 \|\nabla L_t(\mathrm{W}_t)\|_F^2
\end{aligned}
$$

The last term on the right hand side above is bounded as follows:

$$
\begin{aligned}
\eta^2 \|\nabla L_t(\mathrm{W}_t)\|_F^2 &= \eta^2 \|\ell'(y_t g_t(\mathrm{W}_t)) y_t \nabla g_t(\mathrm{W}_t)\|_F^2 \\
&= \eta^2 \left(-\ell'(y_t g_t(\mathrm{W}_t)) \|\nabla g_t(\mathrm{W}_t)\|_F\right)^2 \\
&= \eta^2 Q_t(\mathrm{W}_t)^2 \sum_{r=1}^{m} \|\frac{\partial g_t(\mathrm{W}_t)}{\partial \mathrm{w}_{r,t}}\|^2 \\
&\leq \eta^2 Q_t(\mathrm{W}_t)^2 \quad\quad\quad\quad\quad\quad (\|\frac{\partial g_t(\mathrm{W}_t)}{\partial \mathrm{w}_{r,r}}\| \leq \frac{1}{\sqrt{m}}) \\
&\leq \frac{\eta^2}{\ln(2)} Q_t(\mathrm{W}_t) \quad\quad\quad\quad\quad\quad (Q_t(\cdot) \leq 1/\ln(2)) \\
&\leq \eta Q_t(\mathrm{W}_t) \quad\quad\quad\quad\quad\quad\quad (\text{assumption } \eta \leq \ln(2)) \\
&\leq \eta L_t(\mathrm{W}_t) \quad\quad\quad\quad\quad\quad\quad\quad (Q_t(\cdot) \leq L_t(\cdot))
\end{aligned}
$$

The second term can be bounded as follows:

$$
\begin{aligned}
\langle \nabla L_t(\mathrm{W}_t), \mathrm{W}_t - \mathrm{U} \rangle &= \ell'(y_t g_t(\mathrm{W}_t)) \langle y_t \nabla g_t(\mathrm{W}_t), \mathrm{W}_t - \mathrm{U} \rangle \\
&= \ell'(y_t g_t(\mathrm{W}_t))(y_t g_t(\mathrm{W}_t) - y_t g_t^{(t)}(\mathrm{U})) \quad \text{(Homogeneity, definition of } g_t^{(t)}) \\
&\geq (\ell(y_t g_t(\mathrm{W}_t)) - \ell(y_t g_t^{(t)}(\mathrm{U}))) \quad\quad\quad\quad\quad \text{(convexity of } \ell(\cdot)) \\
&= L_t(\mathrm{W}_t) - L_t^{(t)}(\mathrm{U})
\end{aligned}
$$

Plugging back the above inequalities we get

$$\|W_{t+1} - U\|_F^2 \leq \|W_{t+\frac{1}{2}} - U\|_F^2 \leq \|W_t - U\|_F^2 - 2\eta(L_t(W_t) - L_t^{(t)}(U)) + \eta L_t(W_t)$$
$$= \|W_t - U\|_F^2 - \eta L_t(W_t) + 2\eta L_t^{(t)}(U) \tag{4}$$

Rearranging, dividing both sides by $\eta$, and averaging over iterates we arrive at

$$\frac{1}{T}\sum_{t=1}^{T}L_t(W_t) \leq \sum_{t=1}^{T}\frac{\|W_t - U\|_F^2 - \|W_{t+1} - U\|_F^2}{\eta T} + \frac{2}{T}\sum_{t=1}^{T}L_t^{(t)}(U)$$
$$\leq \frac{\|W_1 - U\|_F^2}{\eta T} + \frac{2}{T}\sum_{t=1}^{T}L_t^{(t)}(U) \qquad \text{(Telescopic sum)}$$

$\square$

**Lemma B.3.** With probability at least $1 - \delta/3$ it holds uniformly over all $t \in [T]$ that $|g_t(W_1)| \leq \sqrt{2\ln(6T/\delta)}$, provided that $m \geq 25\ln(6T/\delta)$.

*Proof of Lemma B.3.* The proof is similar to the proof of Lemma A.1 in Ji and Telgarsky [2019], except for that we have to take into account the randomness due to dropout as well. In particular, there are four different sources of randomness in $g_t(W_1) = g(W_1; x_t, B_t)$: 1) the randomly initialized hidden layer weights $W_1$, 2) the randomly initialized top layer weights a, 3) the input vector $x_t$, $t \in [T]$, and 4) the Bernoulli masks $B_t$, $t \in [T]$. Given input $x_t$ and the dropout mask $B_t$, let $h_t(W) = \frac{1}{\sqrt{m}}B_t\sigma(Wx_t) \in \mathbb{R}^m$ denote the (scaled) output of the dropout layer with hidden weights $W$. It is easy to see that the function $g : W \mapsto \|h_t(W)\|$ is 1-Lipschitz:

$$|g(W) - g(W')| = |\|h_t(W)\| - \|h_t(W')\||$$
$$\leq \|h_t(W) - h_t(W')\| \qquad \text{(Reverse Triangle Inequality)}$$
$$= \sqrt{\sum_{r=1}^{m}(\frac{1}{\sqrt{m}}b_i^{(t)}\sigma(\langle w_{r,1}, x_t\rangle) - \frac{1}{\sqrt{m}}b_i^{(t)}\sigma(\langle w_{r,1}', x_t\rangle))^2}$$
$$= \frac{\sqrt{\sum_{r=1}^{m}(\langle w_{r,1}, x_t\rangle - \langle w_{r,1}', x_t\rangle)^2}}{\sqrt{m}} \qquad \text{(1-Lipschitzness of ReLU)}$$
$$\leq \frac{\sqrt{\sum_{r=1}^{m}\|w_{r,1} - w_{r,1}'\|^2\|x_t\|^2}}{\sqrt{m}} \qquad \text{(Cauchy-Schwarz)}$$
$$= \frac{\|W - W'\|_F}{\sqrt{m}}$$

Using Gaussian concentration (Lemma A.1), we get that $\|h_t(W_1)\| - \mathbb{E}_{W_1}[\|h_t(W_1)\|] \leq \sqrt{\frac{2\ln\left(\frac{6T}{\delta}\right)}{m}}$ with probability at least $1 - \frac{\delta}{6T}$. It also holds that:

$$\mathbb{E}_{W_1}[\|h_t(W_1)\|] \leq \sqrt{\mathbb{E}_{W_1}[\|h_t(W_1)\|^2]}$$
$$= \sqrt{\sum_{r=1}^{m}\mathbb{E}_{w_{r,1}}(\frac{1}{\sqrt{m}}b_{r,t}\sigma(w_{r,1}^\top x_t))^2}$$
$$\leq \sqrt{\frac{\sum_{r=1}^{m}\mathbb{E}_{w_{r,1}}[\sigma(w_{r,1}^\top x_t)^2]}{m}}$$
$$= \sqrt{\mathbb{E}_{z\sim\mathcal{N}(0,1)}[\sigma(z)^2]} = \frac{1}{\sqrt{2}}$$

As a result, we have with probability at least $1 - \frac{\delta}{6T}$ that $\|h_t(W_1)\| \leq \sqrt{\frac{2\ln(6T/\delta)}{m}} + \frac{\sqrt{2}}{2} \leq 1$ whenever $m \geq 25\ln(6T/\delta)$. Now, taking a union bound over all $t \in [T]$, we get that $\|h_t(W_1)\| \leq 1$ holds

simultaneously for all iterates. Conditioned on this event, the random variable $g_t(W_1) = \langle a, h_t(W_1) \rangle$ is zero mean and 1-sub-Gaussian, so that by the general Hoeffding's inequality, for any $t$, with probability at least $1 - \frac{\delta}{6T}$, it holds that $|g_t(W_1)| \leq \sqrt{2 \ln(6T/\delta)}$. Taking union bound over all $t \in [T]$, with probability $1 - \delta/6$ it holds that $|g_t(W_1)| \leq \sqrt{2 \ln(6T/\delta)}$ simultaneously for all $t \in [T]$. Finally, the probability that both of these events hold is no less than $(1 - \delta/6)^2 \geq 1 - \delta/3$, which completes the proof. $\qquad \square$

**Lemma B.4.** Under Assumption 1, for any $\delta \in (0, 1)$, with probability at least $1 - \delta/3$ it holds uniformly for all $t \in [T]$ that:

$$y_t g_t^{(1)}(V) = y_t \langle \nabla g_t(W_1), V \rangle \geq \gamma - \sqrt{\frac{2 \ln(3T/\delta)}{m}}$$

*Proof of Lemma B.4.* By Assumption 1, it holds that $\mathbb{E}_{z,b}[y\langle \psi(z), bx\mathbb{I}\{z^\top x > 0\}\rangle] \geq \gamma$ for all $(x, y)$ in the domain of $\mathcal{D}$. We observe that $y_t g_t^{(1)}(V)$ is an empirical estimate of this quantity:

$$
\begin{aligned}
y_t g_t^{(1)}(V) &= y_t \langle \nabla g_t(W_1), V \rangle \\
&= y_t \sum_{r=1}^{m} \langle \frac{1}{\sqrt{m}} a_r b_{r,t} \mathbb{I}\{x_t^\top w_{r,1} > 0\} x_t, \frac{1}{\sqrt{m}} a_r \psi(w_{r,1}) \rangle \\
&= \frac{1}{m} \sum_{r=1}^{m} y_t \langle \psi(w_{r,1}), b_{r,t} x_t \mathbb{I}\{w_{r,1}^\top x_t > 0\} \rangle
\end{aligned}
$$

For $t, r \in [T] \times [m]$, let $\gamma_{t,r} := y_t \langle \psi(w_{r,1}), b_{r,t} x_t \mathbb{I}\{w_{r,1}^\top x_t > 0\} \rangle$. Note that $\mathbb{E}_{W_1, B_t}[\gamma_{t,r}] = \mathbb{E}_{z,b}[y_t \langle \psi(z), bx_t \mathbb{I}\{z^\top x_t > 0\} \rangle]$. Also, for any $t$, the random variable $\gamma_{t,r}$ is bounded almost surely as follows:

$$|\gamma_{t,r}| \leq |y_t| \, \|\psi(w_{r,1})\| \, |b_{r,t}| \, \|x_t\| \, |\mathbb{I}\{w_{r,1}^\top x_t > 0\}| \leq 1.$$

Therefore by Hoeffding's inequality (Theorem A.2), with probability at least $1 - \frac{\delta}{3T}$, it holds that:

$$y_t g_t^{(1)}(V) - \gamma \geq y_t g_t^{(1)}(V) - \mathbb{E}[y_t g_t^{(1)}(V)] \geq -\sqrt{\frac{2 \ln(3T/\delta)}{m}}$$

Applying a union bound over $t$ finishes the proof. $\qquad \square$

*Proof of Lemma 5.2.* We adopt the proof of Theorem 2.2 in Ji and Telgarsky [2019] for dropout training. Assume that $\|w_{r,t} - w_{r,1}\| \leq \frac{7\lambda}{2\gamma\sqrt{m}}$ holds for the first $T$ iterates of Algorithm 1. Then with probability at least $1 - (\frac{\delta}{3} + \frac{\delta}{3} + \frac{\delta}{3}) = 1 - \delta$, Lemma B.1, Lemma B.3, and Lemma B.4 hold simultaneously. We first prove that $L_t^{(t)}(U) \leq \frac{\lambda^2}{2\eta T}$ for all $t \in [T]$. Using the inequality $\log(1 + z) \leq z$, we get that

$$L_t^{(t)}(U) = \log(1 + e^{-y_t \langle \nabla g_t(W_t), U \rangle}) \leq e^{-y_t \langle \nabla g_t(W_t), U \rangle}$$

To upper-bound the right hand side, we lower-bound $y_t \langle \nabla g_t(W_t), U \rangle$. By definition of $U$, we have

$$y_t \langle \nabla g_t(W_t), U \rangle = y_t \langle \nabla g_t(W_t), W_1 \rangle + \lambda y_t \langle \nabla g_t(W_t), V \rangle \tag{5}$$

We bound each of the terms separately. The first term can be decomposed as follows:

$$
\begin{aligned}
y_t \langle \nabla g_t(W_t), W_1 \rangle &= y_t \langle \nabla g_t(W_1), W_1 \rangle + y_t \langle \nabla g_t(W_t) - \nabla g_t(W_1), W_1 \rangle \\
&\geq -|y_t g_t(W_1)| - |y_t \langle \nabla g_t(W_t) - \nabla g_t(W_1), W_1 \rangle| \tag{6}
\end{aligned}
$$

By Lemma B.3, the first term on right hand side is lower-bounded by $-|g_t(W_1)| \geq -\sqrt{2\ln(6T/\delta)}$. We bound the second term as follows:

$$|y_t\langle\nabla g_t(W_t) - \nabla g_t(W_1), W_1\rangle| = \left|\frac{y_t}{\sqrt{m}}\sum_{r=1}^m a_r b_{r,t}(\mathbb{I}\{w_{r,t}^\top x_t > 0\} - \mathbb{I}\{w_{r,1}^\top x_t > 0\})w_{r,1}^\top x_t\right|$$

$$\leq \frac{1}{\sqrt{m}}\sum_{r\in R_t}|a_r b_{r,t}\langle w_{r,1}, x_t\rangle| \qquad\qquad \text{(Triangle inequality)}$$

$$\leq \frac{1}{\sqrt{m}}\sum_{r\in R_t}|\langle w_{r,t} - w_{r,1}, x_t\rangle| \qquad\qquad (r\in R_t)$$

$$\leq \frac{|R_t|\,\|w_{r,t} - w_{r,1}\|}{\sqrt{m}}$$

$$\leq \frac{49\lambda^2}{4\gamma^2\sqrt{m}} + \sqrt{\frac{49\lambda^2\ln(3T/\delta)}{8\gamma^2 m}} \qquad\qquad \text{(Lemma B.1)}$$

$$\leq \frac{\lambda\gamma}{2} \qquad\qquad\qquad\qquad\qquad\qquad\qquad (7)$$

where the last inequality holds when $m \geq \max\{98\gamma^{-4}\ln(3T/\delta), 2401\gamma^{-6}\lambda^2\} = 2401\gamma^{-6}\lambda^2$. The second term in Equation 5 is bounded as follows:

$$y_t\langle\nabla g_t(W_t), V\rangle = y_t\langle\nabla g_t(W_1), V\rangle + y_t\langle\nabla g_t(W_t) - \nabla g_t(W_1), V\rangle$$

$$\geq y_t\langle\nabla g_t(W_1), V\rangle - |y_t\langle\nabla g_t(W_t) - \nabla g_t(W_1), V\rangle|$$

$$= y_t g_t^{(1)}(V) - \left|\frac{y_t}{\sqrt{m}}\sum_{r=1}^m a_r b_{r,t}(\mathbb{I}\{w_{r,t}^\top x_t > 0\} - \mathbb{I}\{w_{r,1}^\top x_t > 0\})\langle\frac{1}{\sqrt{m}}a_r\psi(w_{r,1}), x_t\rangle\right|$$

$$\geq \gamma - \sqrt{\frac{2\ln(3T/\delta)}{m}} - \frac{1}{m}\sum_{r\in R_t}|a_r b_{r,t}\langle\psi(w_{r,1}), x_t\rangle| \qquad \text{(Lemma B.4)}$$

$$\geq \gamma - \sqrt{\frac{2\ln(3T/\delta)}{m}} - \frac{|R_t|}{m}$$

$$\geq \gamma - \sqrt{\frac{2\ln(3T/\delta)}{m}} - \frac{7\lambda}{2\gamma\sqrt{m}} - \sqrt{\frac{\ln(3T/\delta)}{2m}} \qquad \text{(Lemma B.1)}$$

$$\geq \gamma - \frac{\gamma^2}{7} - \frac{\gamma^2}{14} - \frac{\gamma^2}{14} = \gamma - \frac{2\gamma^2}{7} \geq \frac{5\gamma}{7} \qquad\qquad (8)$$

where the penultimate inequality holds when $m \geq \max\{98\gamma^{-4}\ln(3T/\delta), 2401\gamma^{-6}\lambda^2\} = 2401\gamma^{-6}\lambda^2$. Plugging Equations (7) and (8) in Equation 5, we get that

$$y_t\langle\nabla g_t(W_t), U\rangle \geq -\sqrt{2\ln(6T/\delta)} + \frac{3\lambda\gamma}{14} \geq \ln\left(\frac{2\eta T}{\lambda^2}\right), \qquad (9)$$

where the inequality hold true for $\lambda := 5\gamma^{-1}\ln(2\eta T) + \sqrt{44\gamma^{-2}\ln(6T/\delta)}$. Thus, we have that

$$L_t^{(t)}(U) = \log(1 + e^{-y_t\langle\nabla g_t(W_t), U\rangle}) \leq \frac{\lambda^2}{2\eta T}.$$

We now prove by induction that $\|w_{r,t} - w_{r,1}\| \leq \frac{7\lambda}{2\gamma\sqrt{m}}$ holds throughout dropout training. First, we show that the claim holds for $t = 2$:

$$\|w_{r,2} - w_{r,1}\| = \|\Pi_c(\eta\frac{\partial L_t(B_1 W_1)}{\partial w_{r,1}})\| \leq \|\eta\frac{\partial L_t(B_1 W_1)}{\partial w_{r,1}}\|$$

$$\leq \|\eta\ell'(y_t f_t(B_1 W_1))y_i\frac{\partial f_t(B_1 W_1)}{\partial w_{r,1}}\|$$

$$\leq \frac{\eta}{\ln(2)\sqrt{m}} \leq \frac{7\lambda}{2\gamma\sqrt{m}}, \qquad\qquad (\eta \leq \ln(2))$$

which proves the basic step. Now by induction hypothesis, we assume that the claim holds for all $k \in [t]$, i.e., it holds that $\|w_{r,k} - w_{r,1}\| \leq \frac{7\lambda}{2\gamma\sqrt{m}}$. Therefore, it holds that $\|w_{r,k}\| \leq \|w_{r,1}\| + \|w_{r,k} - w_{r,1}\| \leq c - 1 + 1 \leq c$, where we used the triangle inequality, the fact that $\|w_{r,1}\| \leq c - 1$, and that $m \geq 2401\gamma^{-6}\lambda^2$. This, in particular, means that all iterates $1 < k \leq t$ remain in $\mathcal{W}_c$:

$$W_k = \Pi_c(W_{k-\frac{1}{2}}) = W_{k-\frac{1}{2}} \text{ for all } 1 < k \leq t. \tag{10}$$

For the $t+1$-th iterate, we first upper-bound the distance from initialization in terms of the $Q$ function:

$$\|w_{r,t+1} - w_{r,1}\| = \|\Pi_c(w_{r,t} - \eta\frac{\partial L_t(W_t)}{\partial w_{r,t}}) - w_{r,1}\|$$

$$\leq \|w_{r,t} - \eta\frac{\partial L_t(W_t)}{\partial w_{r,t}} - w_{r,1}\|$$

$$\leq \|\eta\frac{\partial L_t(W_t)}{\partial w_{r,t}}\| + \|w_{r,t} - w_{r,1}\|$$

$$\leq \sum_{k=1}^{t} \|\eta\frac{\partial L_k(W_k)}{\partial w_{r,k}}\|$$

$$\leq \eta\sum_{k=1}^{t} -\ell'(y_k g_k(W_k))\|y_k\frac{\partial g_k(W_k)}{\partial w_{r,k}}\|$$

$$\leq \frac{\eta}{\sqrt{m}}\sum_{k=1}^{t} -\ell'(y_k g_k(W_k))$$

The idea is to turn the right hand side above into a telescopic sum using the identity $W_{k+1} - W_k = W_{k+\frac{1}{2}} - W_k = \eta\ell'(y_k g_k(W_k))y_k\nabla g_k(W_k)$, $k \in [t-1]$. By induction hypothesis, for all $k \in [t]$, Equation (8) guarantees $y_k\langle\nabla g_k(W_k), V\rangle \geq \frac{5\gamma}{7}$. Thus, multiplying the right hand side of (11) by $\frac{7}{5\gamma}y_k\langle\nabla g_k(W_k), V\rangle$, we get that:

$$\|w_{r,t+1} - w_{r,1}\| \leq \frac{7\eta}{5\gamma\sqrt{m}}\sum_{k=1}^{t} -\ell'(y_k g_k(W_k))y_k\langle\nabla g_k(W_k), V\rangle$$

$$= \frac{7}{5\gamma\sqrt{m}}\sum_{k=1}^{t}\langle\eta\nabla L_k(W_k), V\rangle$$

$$= \frac{7}{5\gamma\sqrt{m}}\langle W_{t+\frac{1}{2}} - W_1, V\rangle \qquad \text{(Equation (10))}$$

$$= \frac{7\langle W_{t+\frac{1}{2}} - U, V\rangle + 7\langle U - W_1, V\rangle}{5\gamma\sqrt{m}}$$

$$\leq \frac{7\|W_{t+\frac{1}{2}} - U\|_F\|V\|_F + 7\langle\lambda V, V\rangle}{5\gamma\sqrt{m}} \qquad \text{(Cauchy-Schwarz)}$$

$$\leq \frac{7\|W_{t+\frac{1}{2}} - U\|_F + 7\lambda}{5\gamma\sqrt{m}} \tag{11}$$

Again by induction hypothesis, Equation (4) and Equation (9) hold for all $k \in [t]$, which are used to bound $\|W_{t+\frac{1}{2}} - U\|_F$ as follows:

$$\|W_{t+\frac{1}{2}} - U\|_F^2 \leq \|W_t - U\|_F^2 - 2\eta(L_t(W_t) - L_t^{(t)}(U)) + \eta L_t(W_t) \qquad \text{(Equation (4))}$$

$$\leq \|W_t - U\|_F^2 + 2\eta L_t^{(t)}(U))$$

$$\leq \|W_1 - U\|_F^2 + 2\eta \sum_{k=1}^{t} L_k^{(k)}(U)$$

$$\leq \|\lambda V\|_F^2 + 2\eta t \frac{\lambda^2}{2\eta T} \qquad \text{(Equation 9)}$$

$$\leq \lambda^2 + \frac{\lambda^2 t}{T} \qquad (\|V\|_F \leq 1)$$

$$\leq 2\lambda^2$$

$$\implies \|W_{t+\frac{1}{2}} - U\|_F \leq \sqrt{2}\lambda \qquad (12)$$

Plugging Equation (12) back in Equation (11), we arrive at:

$$\|w_{r,t+1} - w_{r,1}\| \leq \frac{7\sqrt{2}\lambda + 7\lambda}{5\gamma\sqrt{m}} \leq \frac{7\lambda}{2\gamma\sqrt{m}}$$

Which completes the induction step and the proof. $\qquad \square$

A crucial step in giving generalization bounds in expectation via upper-bounding the logistic loss is to control the maximum value the loss can take on any iterate of the algorithm. In particular, we need to upper-bound the instantaneous loss of $g_t^{(t)}(U)$, which appears in the right hand side of Lemma 5.1. To that end, we note that the logistic loss only grows linearly for $z < 0$. More formally, it holds for all $z < 0$ that:

$$\log(1 + e^{-z}) \leq \frac{-z}{\ln(2)} + 1 \qquad (13)$$

as depicted in Figure B.

**Lemma B.5.** Under Algorithm 1, it holds with probability one for all iterates that $L_t^{(t)}(U) \leq \frac{c\sqrt{m}}{\ln(2)} + 1$.

*Proof of Lemma B.5.* Recall that $L_t^{(t)}(U) = \ell(y_t g_t^{(t)}(U))$. First we bound the argument inside the loss function:

$$\left| y_t g_t^{(t)}(W_t) \right| = |y_t \langle \nabla g_t(W_t), U \rangle|$$

$$\leq \sum_{r=1}^{m} \left| \langle \frac{\partial g_t(W_t)}{\partial w_{r,t}}, u_r \rangle \right| \qquad \text{(triangle inequality)}$$

$$\leq \sum_{r=1}^{m} \|\frac{\partial g_t(W_t)}{\partial w_{r,t}}\| \|w_{r,1} + \lambda v_r\| \qquad \text{(Cauchy-Schwarz)}$$

$$\leq \sum_{r=1}^{m} \frac{c - 1 + \lambda/\sqrt{m}}{\sqrt{m}} \qquad (\|w_{r,1}\| \leq c - 1, \|v_r\| \leq 1/\sqrt{m})$$

$$\leq c\sqrt{m} \qquad (\lambda \leq \sqrt{m})$$

Now using Equation (13), we get that

$$L_t^{(t)}(U) = \log(1 + e^{-y_t \langle \nabla g_t(W_t), U \rangle}) \leq \log(1 + \exp(c\sqrt{m})) \leq \frac{c\sqrt{m}}{\ln(2)} + 1.$$

$$\square$$

*Proof of Theorem 4.1.* Note that $W_t$ is conditionally independent from $x_t$ given $x_1, \ldots, x_{t-1}$. Thus,

$$\mathbb{E}_{\mathcal{S}_T}[L_t(W_t)] = \mathbb{E}_{\mathcal{S}_{t-1}}[\mathbb{E}_{(x_t, y_t)} \ell(y_t f_t(W_t)) | \mathcal{S}_{t-1}] = \mathbb{E}_{\mathcal{S}_{t-1}}[L(W_t)]$$

Using the fact that logistic loss upper-bounds the zero-one loss, taking expectation over initialization, taking average over iterates, and using Lemma B.2, we get that:

$$\mathbb{E}_{W_1, a, \mathcal{S}_T}\left[\frac{1}{T}\sum_{t=1}^{T}\mathcal{R}(qW_t)\right] \leq \mathbb{E}_{W_1, a, \mathcal{S}_T}\left[\frac{1}{T}\sum_{t=1}^{T}\ell(y_t f_t(qW_t))\right] \qquad (\mathbb{I}\{z < 0\} \leq \ell(z))$$

$$\leq \mathbb{E}_{W_1, a, \mathcal{S}_T, \mathcal{B}_T}\left[\frac{1}{T}\sum_{t=1}^{T}L_t(W_t)\right] \qquad \text{(Lemma B.2)}$$

$$\leq \frac{\mathbb{E}_{W_1}[\|W_1 - U\|_F^2]}{2\eta T} + \frac{2}{T}\sum_{t=1}^{T}\mathbb{E}_{W_1, a, \mathcal{S}_T, \mathcal{B}_T}[L_t^{(t)}(U)] \qquad \text{(Lemma 5.1)}$$

The first term is upper-bounded by $\frac{\lambda^2}{2\eta T}$ since $\|W_1 - U\|_F^2 = \|W_1 - W_1 - \lambda V\|_F^2 = \lambda^2\|V\|_F^2 \leq \lambda^2$. Bounding the second term is based on the following two facts:

1. By Lemma 5.2, with probability at least $1 - \delta$, it holds that $L_t^{(t)}(U) \leq \frac{\lambda^2}{2\eta T} =: u_1$.

2. By Lemma B.5, it holds with probability one that $L_t^{(t)}(U) \leq \frac{c\sqrt{m}}{\ln(2)} + 1 \leq 2c\sqrt{m} =: u_2$.

Therefore, the expected value of $L_t^{(t)}(U)$ can be upper-bounded as:

$$\mathbb{E}[L_t^{(t)}(U)] \leq (1 - \delta)u_1 + \delta u_2 \leq \frac{\lambda^2}{2\eta T} + 2\delta c\sqrt{m}$$

Choosing $\delta := \frac{1}{4\eta c\sqrt{m}T}$ guarantees that

$$\mathbb{E}[L_t^{(t)}(U)] \leq \frac{\lambda^2}{2\eta T} + \frac{1}{2\eta T} \leq \frac{\lambda^2}{\eta T},$$

where $\lambda := 5\gamma^{-1}\ln(2\eta T) + \sqrt{44\gamma^{-2}\ln(24\eta c\sqrt{m}T^2)}$. $\qquad\qquad\square$

*Proof of Theorem 4.2.* First, recall the following property of the logistic loss:

$$\mathbb{I}\{z < 0\} \leq -2\ln(2)\,\ell'(z) \leq 2\ln(2)\,\ell(z)$$

which implies that $\mathcal{R}(W_t; B_t) \leq 2\ln(2)\,Q(W_t; B_t)$, where $Q(W; B) := \mathbb{E}_{\mathcal{D}}[-\ell'(yg(W; x, B)]$ is the expected value of the negative derivative of the logistic loss. On the other hand, taking the empirical average over the training data, and using Lemma 5.1 and Lemma 5.2, we conclude that the following holds with probability at least $1 - \delta$:

$$\frac{1}{T}\sum_{t=1}^{T}Q_t(W_t) \leq \frac{1}{T}\sum_{t=1}^{T}L_t(W_t)$$

$$\leq \frac{\|W_1 - U\|_F^2}{\eta T} + \frac{2}{T}\sum_{t=1}^{T}L_t^{(t)}(U) \qquad \text{(Lemma 5.1)}$$

$$\leq \frac{\lambda^2}{\eta T} + \frac{2}{T}\sum_{t=1}^{T}\frac{\lambda^2}{2\eta T} \qquad \text{(Lemma 5.2)}$$

$$\leq \frac{2\lambda^2}{\eta T}.$$

Given the dropout masks $\mathcal{B}_T$, since $Q(W_t; B_t) = \mathbb{E}_{\mathcal{D}}[Q_t(W_t)]$, we know that $\sum_{t=1}^{T}Q(W_t; B_t) - \sum_{t=1}^{T}Q_t(W_t)$ is a martingale difference with respect to the past observations, $\mathcal{S}_{T-1}$. We next show that the average of $Q_t(W_t)$ on the right hand side above is close to the average of $Q(W_t; B_t)$, using

Theorem A.3, similar to Lemma 4.3. of Ji and Telgarsky [2019]. First, this martingale difference sequence is bounded almost surely as $R := 1/\ln(2) \geq Q(W_t; B_t) - Q_t(W_t)$, simply because $0 \leq -\ell'(z) \leq 1/\ln(2)$. The conditional variance can be bounded as:

$$
\begin{aligned}
V_t &:= \sum_{t=1}^{T} \mathbb{E}[(Q(W_t; B_t) - Q_t(W_t))^2 | \mathcal{S}_{t-1}] \\
&= \sum_{t=1}^{T} Q(W_t; B_t)^2 - 2Q(W_t; B_t)\mathbb{E}[Q_t(W_t)|\mathcal{S}_{t-1}] + \mathbb{E}[Q_t(W_t)^2|\mathcal{S}_{t-1}] \\
&\leq \sum_{t=1}^{T} \mathbb{E}[Q_t(W_t)^2|\mathcal{S}_{t-1}] \qquad\qquad\qquad (\mathbb{E}[Q_t(W_t)|\mathcal{S}_{t-1}] = Q(W_t; B_t)) \\
&\leq \frac{1}{\ln(2)} \sum_{t=1}^{T} \mathbb{E}[Q_t(W_t)|\mathcal{S}_{t-1}] \qquad\qquad (0 \leq Q_t(W_t) \leq 1/\ln(2)) \\
&= \frac{1}{\ln(2)} \sum_{t=1}^{T} Q(W_t; B_t)
\end{aligned}
$$

Thus, using Theorem A.3 with $R \leq 1/\ln(2)$ and $V_t \leq \sum_{t=1}^{T} Q(W_t; B_t)/\ln(2)$, we conclude that with probability at least $1 - \delta$ it holds that

$$
\sum_{t=1}^{T} Q(W_t; B_t) - \sum_{t=1}^{T} Q_t(W_t) \leq (e-2) \sum_{t=1}^{T} Q(W_t; B_t) + \frac{\ln(1/\delta)}{\ln(2)}
$$

$$
\implies \frac{1}{T} \sum_{t=1}^{T} Q(W_t; B_t) \leq \frac{4}{T} \sum_{t=1}^{T} Q_t(W_t) + \frac{4\log(1/\delta)}{T}
$$

Plugging the above back in $\mathcal{R}(W_t; B_t) \leq 2\ln(2) Q(W_t; B_t)$, and averaging over iterates we have:

$$
\frac{1}{T} \sum_{t=1}^{T} \mathcal{R}(W_t; B_t) \leq \frac{16\ln(2)\lambda^2}{\eta T} + \frac{8\ln(2)\ln(1/\delta)}{T}
$$

which completes the proof. $\qquad\qquad\qquad\qquad\qquad\qquad\qquad\qquad\qquad\qquad\qquad\qquad\qquad\qquad$ $\square$