[Reviews · NeurIPS 2020]

Review 1

Summary and Contributions: This paper gives test error guarantees for training a two-layer network with ReLU activations and dropout as an explicit regularizer. The proof extends the work of Ji and Telgarsky (2019) that uses the NTK approximation and is based on a margin assumption in the NTK feature space.

Strengths: This work studies an important problem of understanding the theoretical properties of dropout. The paper is mostly clearly written and most of the proofs in the paper seem correct.

Weaknesses: I think there is a major technical mistake in the paper. To prove their main result, the authors rely on an inequality presented after line 259 which says that the expected zero-one *test* error is at most the expected *training* error. This should not be correct because the test error is usually higher than the training error. There is no generalization analysis in the paper, e.g., using Rademacher complexity as in Ji and Telgarsky (2019). The proof of this inequality is given after line 561 in the supplementary, but there is no explanation and it seems incorrect. Am I missing something? Another issue with the paper is that it seems that the guarantees hold for the initialization and only "worsen" with the gradient updates. Therefore, if we consider the algorithm which initializes W_1 and then satisfies W_t = W_1 for all t, it has the same guarantees as SGD. Is this correct? Where in the proofs does SGD have an advantage over the latter algorithm? If it does not, then this is a clear limitation of the result and it does not explain the benefits of training with SGD. In the discussion, the authors claim that following Nagarajan et al., implicit bias may not explain the generalization of SGD. I think that this statement is incorrect. Nagarajan et al. show that uniform convergence bounds may not explain generalization, this is not the same. The implicit bias of SGD implies good generalization (because this is what we see in practice), and it may be possible to explain it and derive generalization guarantees by other methods (not uniform convergence bounds). Other minor comments: 1. I think it would be good to emphasize the differences between the results on SGD with dropout and SGD without dropout. We should expect better guarantees for dropout because it generalizes better in practice. Is this the case in this paper? 2. In Page 8 it is not clear which Lemma is used to prove which Lemma. This should be written more clearly. 3. In Algorithm 1, what is p? 4. Typos: throughout the paper miss-classification should be misclassification. Line 122: "min" is missing. Line 271: ">0" is missing.

Correctness: There seems to be a major mistake in the proof of the main result. See weaknesses above.

Clarity: Yes

Relation to Prior Work: Yes

Reproducibility: Yes

Additional Feedback: -------------- Post author response --------------------- I have read the response, other reviews and relevant parts of the paper again. There is no technical mistake. Thank you for clarifying my misunderstanding. I updated the score.


Review 2

Summary and Contributions: I have read the authors' response and appreciate their explanation. I improve my score but it would be nice if the authors include the clarifications as response to my comments in the final version. ============================================================= This work studies convergence rates of Dropout guaranteeing epsilon-suboptimality in the test error. In addition, it shows that dropout implicitly compresses the network.

Strengths: As far as I know, there has not been any work that studies properties of dropout in such theoretical settings.

Weaknesses: I have to admit that I am not qualified to follow all theoretical results in this paper. However, it could have been easier for someone like me to follow if the insights from each theoretical result were discussed more and maybe even add a toy example for motivation.

Correctness: -I would appreciate if the authors had discussed iteration complexity in practice. We know that applying dropout in neural networks come with a negligible cost because it is just a matter of masking weights. - The logistic loss equation in line 123 is wrong. Also, the loss function \ell takes two parameters in line 122 and one parameter in line 123. - As I mentioned, I was not able to follow the derivation of the theoretical results but I wonder if the main reason for those results is the max-norm instead of dropout.

Clarity: - I find paragraph in lines 148-151 rather abrupt. It would have been nice if the authors gave a bit background on how NTK is related to this problem. - I cannot follow the notation in lines 167-169. What is k? Is it iteration number? What do you mean by linearization? Why \nabla g_t is called features? - In basic facts, the claim in line 174 is not obvious to me but it is also because I was not able to follow the notation. - The misclassification error used in theorem 4.3 is confusing. \mathcal{R} takes all weights of the neural network as inputs. Why does it take qW_t in this theorem?

Relation to Prior Work: The authors discussed prior work in detail.

Reproducibility: No

Additional Feedback: - The paper focuses on a binary classification problem. I would like to know what happens for more complicated problems and networks.


Review 3

Summary and Contributions: The paper studies the convergence and compression properties of neural networks trained with dropouts. They make two important and novel contributions. 1. Convergence rates (non-asymptotic) for two layer neural networks with ReLU activation units. 2. Compression - They show that dropout supports compression. In particular, there exists a sub-network that can generalize as well as the whole network. This has interesting relations to the lottery ticket hypothesis.

Strengths: The paper tackles a very interesting problem which is not well understood in the deep learning community. This should be interesting to the nips community.

Weaknesses: In Assumption 1, (q, gamma)-margin: \phi (line 185) is not defined. Also, since assumption 1 is crucial to the proof and statement of all the results in the paper. It might have been helpful, to have more discussion around some of the questions below. why is assumption 1 necessary or why do we expect this to hold on real datasets or some example datasets where such a property holds.

Correctness: The claims seem to be correct.

Clarity: Yes.

Relation to Prior Work: Yes.

Reproducibility: Yes

Additional Feedback: I have read the rebuttal and other reviews. I will keep my score unchanged.


Review 4

Summary and Contributions: This paper gives a proof on the dropout sub-optimalities on the two layer RELU neural networks. Assumptions of the analysis include only the first layer is trainable, overparameterization and data separation.

Strengths: This draft gives a proof on the convergence of the dropout training for the two layer neural overparameterized network. In particular given some margin assumptions, the mis-classification error decreases with a speed of roughly O(1/T).

Weaknesses: The work looks fine to me. I understand the assumption that only the first layer is trainable is used a lot in previous work. It will be better if we can make both layers trainable.

Correctness: Overall the arguments look fine to me, but I did not go through the details.

Clarity: the draft is written fine.

Relation to Prior Work: yes the references seem adequate to me.

Reproducibility: Yes

Additional Feedback: I suggest we ignore the reproducibility question for the theory paper. ================ I have read the authors' response. I will keep my ratings unchanged.

[Author Response · NeurIPS 2020]

We thank the reviewers for taking the time to read and comment on our submission despite these challenging times.

**Reviewer #1.**   You wrote: "This work studies an important problem . . . The paper is mostly clearly written and most
of the proofs in the paper seem correct." Thanks for acknowledging the strengths of our submission.

• *Regarding major technical mistake*. There is no mistake here. We are simply using the fact that the zero-one loss is
bounded from above by the logistic loss, and taking expectations on both sides. This is what gives the inequality after
line 259. What it states is that the expected zero-one loss, i.e., the probability of misclassification, is bounded by
expected logistic loss; there is no *training error* involved in this expression. A detailed derivation is in the Appendix
(look for Proof of Theorem 4.3 on lines 559-561).

• *No generalization analysis*. Theorems 4.3 and 4.6 are our generalization error bounds. Instead of analyzing the
Rademacher complexity, we use a martingale concentration inequality to bound the generalization error. Please see
the Proof of Theorem 4.6. in the appendix for more details.

• Our generalization guarantees do not *worsen* with the gradient updates, they *improve*. In both Theorem 4.3 as well as
Theorem 4.6, we bound the generalization error by $\tilde{\mathcal{O}}(1/T)$, which implies that the generalization error decreases
with $T$, i.e., with the number of gradient updates in Algorithm 1.

• The constant strategy $W_t = W_1$ would incur a constant error at each iteration, and thus not decay with $T$. In sharp
contrast, the error of the iterates of Algorithm 1 decays as $\tilde{\mathcal{O}}(1/T)$. The advantage of Algorithm 1 over the constant
strategy shows up when we apply Lemma 5.4 to the right hand side of the inequality given by Lemma 5.3.

• Thanks for your comment on Nagarajan and Kolter – we will make the *uniform convergence* discussion explicit in the
final version of our paper to better convey the following quote from their paper: ". . . uniform convergence provably
cannot "explain generalization" – even if we take into account the implicit bias of GD to the fullest extent possible."

• Regarding your minor comments, 1) we will extend the language in Remark 4.5 to emphasize on computational
learning theoretic advantages of dropout over SGD; 2) we will make it more clear and precise; 3-4) thanks for
catching the typos, $p$ should be $q$.

We hope that our response has clarified all of your technical concerns, in which case, please reconsider your score.

**Reviewer #2.**   Thanks for acknowledging the novelty of our results.

• The definition of logistic loss on line 123 is not wrong; feel free to refer to any ML book. You can also refer to the
prior work, e.g., [Cao and Gu, 2019], [Ji and Telgarsky, 2019], etc. It is typical to define a loss function as $\ell(y, f(x))$
and overload the notation to write $\ell(z) = \ell(yf(x))$; e.g., for logistic loss this would be $\log(1 + \exp(-yf(x)))$.
Please see the definition of hinge loss for another example.

• As we state in Remark 4.4, with high probability, dropout iterates will not violate the max-norm constraints, i.e.,
these constraints are not active in a typical run of dropout whatsoever. We only need the max-norm constraints to
give generalization error bounds in expectation.

• Yes, $k \in \{1, \ldots, T\}$ represents an iterate. Linearization means we approximate a function with its best linear
approximation, given in terms of the gradient of the function at that point. This is what is happening on lines 167-169.
Neural tangent kernel (NTK) analysis is based on using $\nabla g_t$ as features; see the preliminaries on lines $141 - 151$.

   • Proof of basic fact on line 174:
$$\langle \nabla g_t(W_t), W_t \rangle = \sum_{r=1}^{m} \langle \frac{1}{\sqrt{m}} a_r b_{r,t} \mathbb{I}\{w_{r,t}^\top x_t \geq 0\} x_t, w_{r,t} \rangle = \frac{1}{\sqrt{m}} \sum_{r=1}^{m} a_r b_{r,t} \sigma(w_{r,t}^\top x_t) = \frac{1}{\sqrt{m}} a^\top B_t \sigma(W_t x_t) = g_t(W_t)$$

• The weights returned by the algorithm are scaled by $q$ to account for the dropout. Hence, the risk takes the argument
$qW_t$. We make the following remark on lines $139 - 140$: "at test time, the weights are multiplied by $q$ so as to make
sure that the output at test time is on par with the expected output at training time."

We hope that our response has clarified all of your technical concerns, in which case, please reconsider your score.

**Reviewer #3.**   Thanks for your encouraging comments! We define the *neural tangent feature* $\phi_x : z \mapsto x\mathbb{I}\{z^\top x > 0\}$
on line 150 after Eq. 1. Assumption 1 is simply a margin assumption in the RKHS of the NTK. It is a mild and
reasonable assumption that the data becomes linearly separable after mapping it into a high-dimensional non-linear
feature space; this idea has been the cornerstone of kernel methods using the RBF kernel, for example, and for learning
with neural networks. Further, this assumption yields bounds under mild over-parametrization compared to other works.

We hope that our response has clarified all of your technical concerns, in which case, please reconsider your score.

**Reviewer #4**   Thanks for your positive review!

[Meta-Review · NeurIPS 2020]

The work presents a non-asymptotic rate of convergence on test error via drop-out for 2-layer ReLU networks in the NTK regime. The reviewers appreciate the results, the presentation, and the coverage of the related work. Reviewers felt that additional discussions on the assumptions, e.g., the margin assumption, training of only hidden layer, etc., would enrich the paper.